# Challenges and Possibilities of Archaeological Sites Virtual Tours: The Ulaca *Oppidum* (Central Spain) as a Case Study

Miguel Ángel Maté-González [1,2,3,4,*], Jesús Rodríguez-Hernández [5], Cristina Sáez Blázquez [2,3,6], Libertad Troitiño Torralba [3,7], Luis Javier Sánchez-Aparicio [2,3,8], Jesús Fernández Hernández [2], Tomás Ramón Herrero Tejedor [9], José Francisco Fabián García [10], Marco Piras [4], Carlos Díaz-Sánchez [5], Diego González-Aguilera [2,3], Gonzalo Ruiz Zapatero [5] and Jesús R. Álvarez-Sanchís [5]

1   Department of Topographic and Cartography Engineering, Escuela Técnica Superior de Ingenieros en Topografía, Geodesia y Cartografía, Universidad Politécnica de Madrid, Mercator 2, 28031 Madrid, Spain
2   Department of Cartographic and Land Engineering, Higher Polytechnic School of Ávila, Universidad de Salamanca, Hornos Caleros 50, 05003 Ávila, Spain; u107596@usal.es (C.S.B.); lj.sanchez@upm.es (L.J.S.-A.); j.f.h@usal.es (J.F.H.); daguilera@usal.es (D.G.-A.)
3   Gran Duque de Alba Institution, Diputación Provincial de Ávila, Paseo Dos de Mayo, 8, 05001 Ávila, Spain; ltroitin@ucm.es
4   Department of Environment, Land and Infrastructure Engineering, Politecnico di Torino, 10129 Torino, Italy; marco.piras@polito.it
5   Department of Prehistory, Ancient History and Archaeology, Complutense University of Madrid, Prof. Aranguren s/n, 28040 Madrid, Spain; jesusrodriguez@ucm.es (J.R.-H.); cardia01@ucm.es (C.D.-S.); gonzalor@ghis.ucm.es (G.R.Z.); jralvare@ghis.ucm.es (J.R.Á.-S.)
6   Department of Electric, System and Automatic Engineering, Universidad de León, Campus Vegazana, s/n, 24007 León, Spain
7   Department of Human Geography, Complutense University of Madrid, Prof. Aranguren s/n, 28040 Madrid, Spain
8   Department of Construction and Architectural Technology, Escuela Técnica Superior de Arquitectura de Madrid, Universidad Politécnica de Madrid, Avda. Juan de Herrera 4, 28040 Madrid, Spain
9   Department of Agroforestry Engineering, Escuela Técnica Superior de Ingeniería Agronómica, Alimentaria y de Biosistemas, Universidad Politécnica de Madrid, Campus Ciudad Universitaria, Av. Puerta de Hierro, nº 2–4, 28040 Madrid, Spain; tomas.herrero.tejedor@upm.es
10  Servicio Territorial de Cultura de Ávila, Junta de Castilla y León, Plaza Fuente el Sol, s/n, 05001 Ávila, Spain; fabgarfr@jcyl.es
*   Correspondence: mategonzalez@usal.es or miguelangel.mate@upm.es; Tel.: +34-920-35-35-00

**Abstract:** This research presents a virtual tour performed on the *oppidum* of Ulaca, one of the most relevant archaeological sites of the Iberian Peninsula during the Late Iron Age (*ca.* 400–50 BC). Beyond the clear benefits of the tool to the interpretation, dissemination, and knowledge of the mentioned archaeological site and its surroundings, the novelty of this research is the implementation of the platform in alternative scenarios and purposes. In this way, the present work verifies how the access to multi-source and spatially geolocated information in the same tool (working as a geospatial database) allows the promotion of cross-sectional investigations in which different specialists intervene. This peculiarity is also considered useful to promote tourism with an interest beyond the purely historical/archaeological side. Likewise, the possibility of storing and managing a large amount of information in different formats facilitates the investigation in the contexts of excavations and archaeological or environmental works. In this sense, the use of this kind of tool for the study of cultural landscapes is especially novel. In order to better contextualize the potential of the virtual tour presented here, an analysis about the challenges and possibilities of implementing this tool in environments such as the Ulaca *oppidum* is performed. The selected site stands out for: (i) being in a unique geological, environmental and ecological context, allowing us to appreciate how human beings have modified the landscape over time; (ii) presenting numerous visible archaeological remains with certain conservation problems; and (iii) not having easy access for visitors.

**Keywords:** *oppidum*; vettones; virtual tour; cultural heritage; cultural landscapes

## 1. Introduction

Territory has progressively assimilated its condition as heritage resource in recent decades [1,2], leading to a rapprochement between territorial and heritage assessments and interpretations. These conceptual changes have made it possible to broaden the simplified vision usually associated with heritage, moving from the monument to the territory [3] and vice versa. In other words, the concept of the territorial context as a heritage resource allows determining, understanding, and valuing the unique pieces on which a place is structured.

Territorial heritage integrates environmental, cultural, social, and economic functions, in which tourism activity has gained special prominence in recent years, and especially, in inland or natural territories, as a consequence of the pandemic generated by COVID-19. The heritage valuation process that the territory is going through is contributing to overcome the idea of heritage as a burden and to be considered as a fundamental resource in the most innovative territorial development strategies. An integrated heritage analysis implies discovering that the territory is the depository of numerous resources, both material and intangible, natural and cultural, which represent not only the signs of its past identity but are also the key to base future development [4,5]. However, heritage cannot be managed apart from the processes of construction and social appreciation of the territory as cultural capital [6]. That is why, within the framework of development approaches on a human scale [7], territory, culture, heritage, and economy must be managed under unitary criteria [8]. Troitiño [9] pointed out that the organization and management of heritage territories, considering the uniqueness of the territories in which they were located, must be capable of channeling new functionalities, such as tourism, leisure, cultural, landscape or environmental activities, as well as assigning them a clear and differentiated role in current urban and territorial structures. It is time to face the potentialities, problems, effects, and consequences related to the valorization of heritage, calling for a heritage planning and management of the territory that offers a transversal, dynamic, and integrating vision [10], and that considers the social, cultural, economic, environmental, and functional dimensions [11]. It is therefore required to overcome a plan focused, in some cases, on the passive protection of heritage and, in others, on the promotion or design of "tourist territories" that mask a real estate overproduction or speculative processes [12].

The preservation of heritage is one of the aspects that every advanced society must face in an unavoidable way. Each community, taking into account its collective memory and past, is responsible for the identification and management of its heritage [13]. The latest International Charter of Restoration (Charter of Krakow, 2000) [14] echoes the need to persist in plans and actions for the training and education of citizens that promotes the dissemination and importance of heritage and its conservation. In this context, it is necessary for educational policies to integrate these actions into national education systems at different levels. This claim is included in most of the recommendations of international organizations (UNESCO, ICOMOS, ICCROM, OMT...). However, making a correct dissemination of heritage is not an easy task. There are currently numerous initiatives that try to offer society means of transmitting knowledge about heritage, but the possibility of not being as useful, attractive, or efficient as intended must be considered [15]. As a rule, there are different difficulties when understanding and interpreting the historical meaning of heritage by society as a non-specialized public [15]. Apart from the general information that can be obtained about a heritage asset, a tradition, or a cultural landscape, it is associated with a large amount of historical information that is not always easy to transmit [16].

In recent decades, work has been done to promote and value the heritage that surrounds us, with the aim of obtaining tools close to citizens. The goal is to facilitate greater access to culture for all sectors of the population and increase the educational levels of society. Thus, as a result of the advances experienced by Geoinformation technologies, several initiatives try to use methods of visualization and virtualization of cultural heritage for these purposes [17–21]. These applications are presented as an exceptional instrument capable of allowing users to interact with heritage through digital tools [22].

This so-called Virtual Archaeology is openly considered as a scientific discipline, collected in the London Charter (2009) [23] and the Seville Principles (2011) [24], since these works are the result of the research and the development of systems and ways of virtualization for a better understanding of cultural heritage [24]. These tools are having a great impact on the development and practice of tourism, not only in terms of supply, but also demand. Digital resources and the generation of apps have multiplied considerably in the last decade, especially since the beginning of the pandemic in the first quarter of 2020 [25]. The restrictions applied to mobility have implied a reaction from the destinations to keep their offer active, to bring the visitor closer to all the attractions from home, with the aim of generating expectations that promote the intention of moving towards these territories. In this way, users, who are increasingly accustomed to the use of new technologies, have the possibility of accessing information that is not easy to transmit (because they are not physically present, such as intangible cultural heritage or assets or traditions that no longer exist, sites with difficult access or in a remote place, among other circumstances), thus enriching the activity carried out [26–28].

Under the bases previously shown, the research project ULACA <sup>VIRTUAL TOUR</sup> was proposed. The mentioned work is focused on the development of a novel web platform for the valorization of the Ulaca *oppidum* (located in the Spanish province of Ávila), being in line with the sustainable tourism promoted by the 2030 Agenda for Sustainable Development [29]. This platform focuses on the tourism of this archaeological site, but also on the researchers who participate in the different studies on pre-Roman civilizations and the students coming from different levels of education, especially disciplines related to compulsory education and university careers in preservation and enhancement of cultural heritage. To this end, an efficient combination of 360 images, 3D models, and a geospatial database using several web-programming languages (HTML-5, CSS-3, Php, JavaScript and XML) was used.

As can be observed, Section 2 describes the case under study, as well as the number of visitors received by Ulaca and other *oppida* in the province of Ávila. Section 3 presents the methodology used to generate the virtual tour, while Section 4 presents the principal results. Likewise, in Section 5, the discussion is developed and, finally, in Section 6 the main conclusions obtained in this work are presented.

## 2. Ulaca *Oppidum* and the Celtic Archaeotourism in the Region of Ávila

The province of Ávila, its history, culture, and society are reflected in its heritage, its historical and artistic legacy and traditions of past times and cultures [30,31]. Its landscapes are the reflection of the vicissitudes of life and traditions of its current and past inhabitants, whose traces are visible in important monuments and in traditions that are still preserved today [32]. It is an extensive territory with an area of 8049 km$^2$ and a current population of 171,265 inhabitants, unevenly distributed in 248 municipalities, which translates into a population density of 21.28 inhabitants/km$^2$, being the ninth province with the lowest ratio inhab/km$^2$ of Spain. There is a clear predominance in the region of small municipalities (less than 500 inhabitants), representing 82% of the province. This proportionality is inverted if the variable used is the population concentration, where Ávila capital shows its supremacy, concentrating 34.40% of the population.

This previous inequality is an inherent condition not only in the territorial reality, but also at the level of the existing tourist and cultural appreciation. Ávila has an unquestionable heritage diversity, both from the cultural, natural, and immaterial point of view. At the international level, the province highlights for the UNESCO World Heritage declaration of "Old City of Ávila and Churches outside the walls" (1985). But beyond the existing values in the capital, the rest of the Ávila municipalities hide numerous heritage elements and a relevant cultural offer, such as the *oppida* of La Mesa de Miranda (Chamartín), Las Cogotas (Cardeñosa), El Raso (Candeleda), and Ulaca (Solosancho) (Figure 1). However, the reality of these resources shows that not all of them are adequate to function as "tourist products".

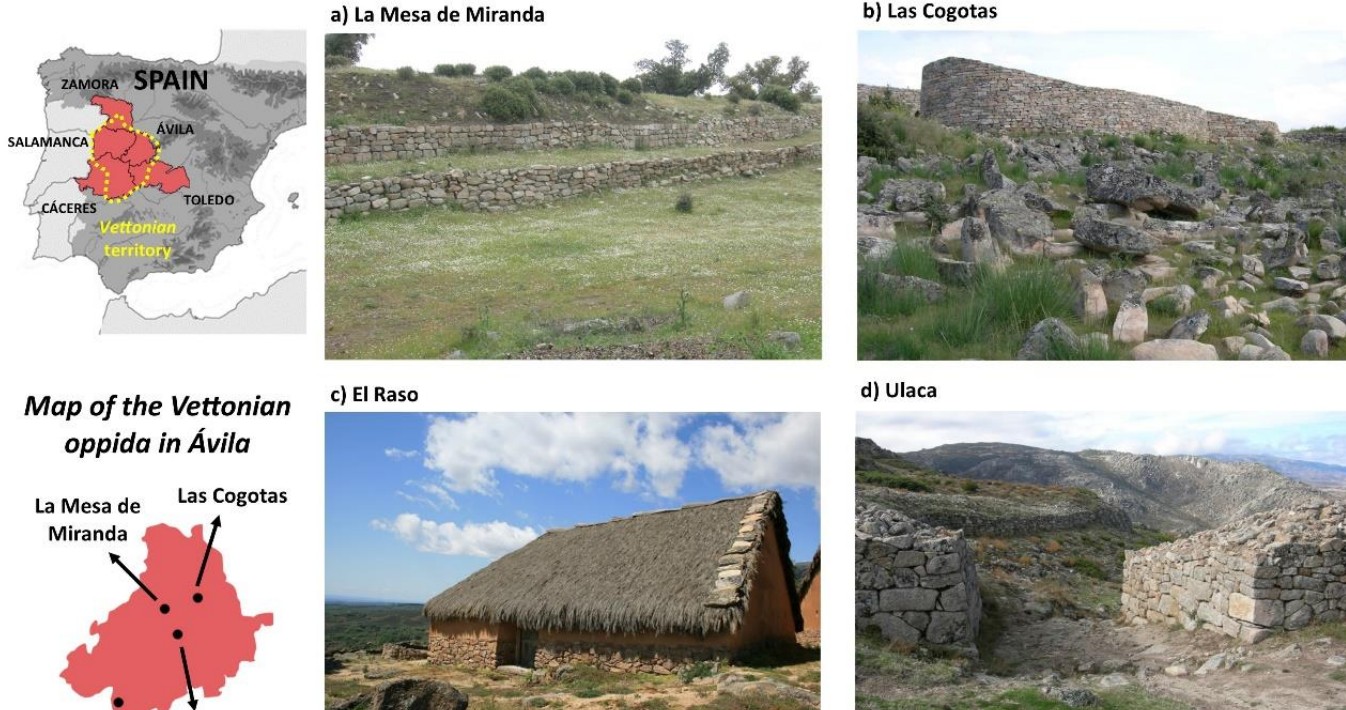

**Figure 1.** Main Vettonian *oppida* in Ávila: (**a**) La Mesa de Miranda, (**b**) Las Cogotas, (**c**) El Raso and (**d**) Ulaca.

The crisis of the historical model of organization and management of the territory, as well as the deep change of the traditional economic pillars of rural and mountain areas in the last three decades, has led to certain difficulties in finding new uses and functions. In this regard, tourism has been one of the most relevant economic activities on the way to the search for a new development model in these areas.

Tourism development must contemplate territorial, economic, social, cultural, and environmental dimensions and must have a multifunctional nature, in which agricultural, touristic, industrial, and environmental activities and different services are integrated, thus falling into the so-called functional monocultures is avoided. It is therefore essential to prepare the territory and society for a future project. Setting up tourist territories implies understanding the system code and identifying the reference elements that contribute to the generation of visitor flows [13].

Articulating territories for tourism (such as the case of Ávila) requires complementarity among the tourist practices that visitors can carry out in them, in the so-called archaeotourism [33]. Certainly, this modality is not the main motivation for tourists and hikers to travel to the region, but in recent years (ignoring the unique situation of a pandemic), it has generated an increasing interest. Between 2015 and 2019, *oppida* available for public visits have received about 92,000 visitors, highlighting the *oppidum* of El Raso and Ulaca that concentrate more than 82% of the total visits (62.12% and 20.70% respectively). These values are approximate, since the guards who collect the influx of visitors do not work every day and there have occasionally been periods without guards (Table 1 and Figure 2).

**Table 1.** Information about the visitors of the *oppida*, source: Territorial Service of Culture of Ávila. Traveler data, source: Rural Tourism (RT) Accommodation Survey. INE (National Statistical Institute of Spain) 2015–2020.

|  | El Raso | Las Cogotas | La Mesa de Miranda | Ulaca | Travelers RT |
|---|---|---|---|---|---|
| 2015 | 13,042 | 897 | 1679 | 2920 | 114,939 |
| 2016 | 12,975 | 892 | 1436 | 5253 | 114,132 |
| 2017 | 11,371 | 1597 | 1612 | 2480 | 123,972 |
| 2018 | 8103 | 1979 | 1432 | 2827 | 148,418 |
| 2019 | 8118 | 867 | 1238 | 2846 | 133,666 |
| 2020 | 3148 | 894 | 1167 | 2590 | 56,860 |

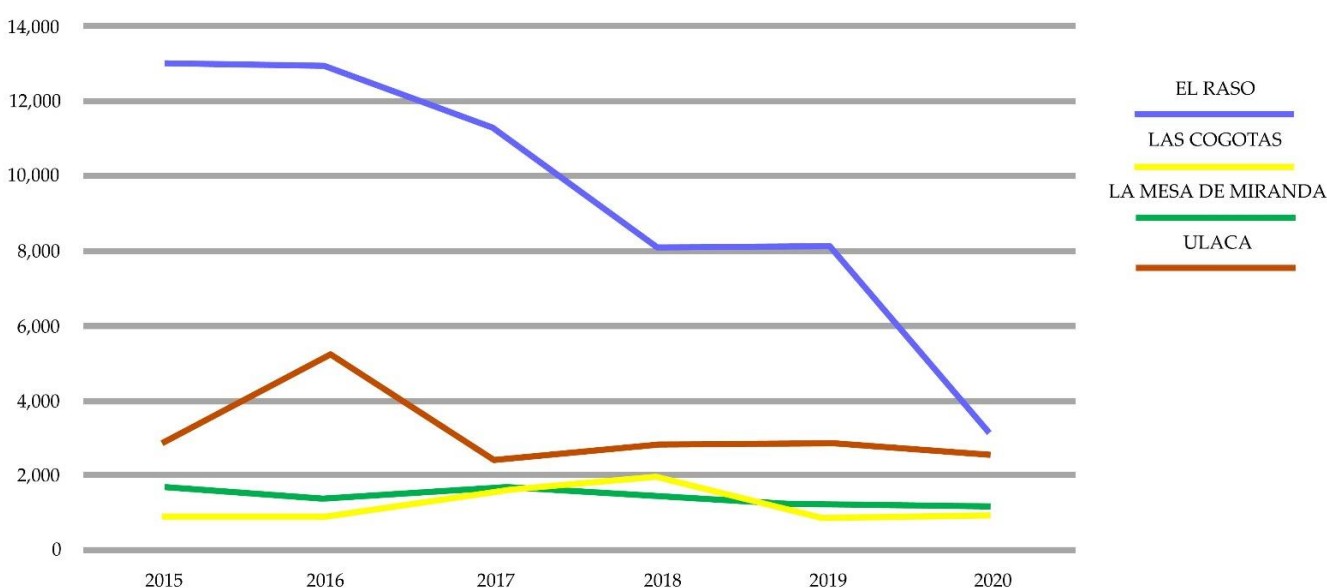

**Figure 2.** Graph about the dynamics of visitors to the *oppida* of Ávila (2015–2020). Source: Territorial Service of Culture of Ávila.

These types of resources are undoubtedly attractive; however, their tourist and heritage values are deficient and tools such as Virtual Tours, whose pilot experience in the Ulaca *oppidum* is a new milestone in the province, can be key for a greater dissemination of their potential.

Destinations, given the strong competition within the tourism market, work not only to improve the situation of the "traditional resources" of attraction, but also generate new products and design strategies. These last are articulated around an offer that must be based on differentiation in order to diversify the profile and demand flows, so conditioned in the case of Avila (weekend visitors or short vacation periods, who carry out tourist practices typical of rural and nature tourism, or the so-called return tourists).

Regional and local institutions, to promote the offer, are supporting minority tourism modalities, such as ornitourism, mycotourism, or star and stellar tourism [34]. In fact, considering the number of arrivals of travelers to rural tourism establishments recorded in 2019—133,666—and relating them to the total number of visitors to the *oppida* in the same year, the latter barely account for 10%. Therefore, it would be advisable to work on the development of archaeological tourism of Ávila as a strategic line, not exclusively from the tourist perspective, but focused on territorial development.

In this regard, the founding milestone of archaeotourism in the province of Ávila can be found in the celebration of the successful exhibition "Celtas y Vettones" (Celts and Vettones) [35], during 2001, which was visited by about 120,000 people and was an authentic cultural and tourist event in the city [36] (p. 407). This initial impulse was consolidated with the installation in the headquarters of the Ávila Provincial Council of the

permanent exhibition "*Vettonia. Cultura y Naturaleza*" (Vettonia. Culture and Nature) [37]. Another key element in the dissemination of the heritage bequeathed by the protohistoric communities has been the restoration and adaptation for the visit of some of the main sites of the Iron Age of Ávila and Salamanca within the framework of the European project INTERREG III-A (2003–2005) [38]. Among these actions, the creation of archaeological visitor centers, the signposting of the access route, the musealization of the surroundings, the cleaning and reconstruction of the defenses, especially around the gates (Figure 1), can be highlighted. In this way, a tourist route has been consolidated around these sites and other important settlements in northern Portugal [39]. In addition, around the main pre-Roman sites in Ávila, several festivals and "Celtic-Vetton" markets have arisen, such as the Celtic Moon in Ulaca, with fifteen editions [40], Lugnasad in La Mesa de Miranda with fourteen editions [41], or the Celtic festival in El Raso with nine editions.

*Ulaca Oppidum*

The Vettones were one of the most prominent people in Celtic Hispania [42]. According to classical Greco–Latin authors, they occupied a wide territory of the Western Plateau corresponding to the current provinces of Ávila, Salamanca, and part of Toledo, Zamora, and Cáceres [43–46]. Throughout the Late Iron Age (*ca*. 400–50 BC) the first cities or *oppida* arose in this area of the western Peninsula. These nuclei controlled a wide territory to ensure the subsistence of their hundreds or thousands of inhabitants and they were usually built in strategic places that had considerable natural defenses. Even so, settlements were reinforced with imposing stone walls, ditches, and *chevaux-de-frise* (Figure 1) [47,48]. In the vicinity of the *oppida* there were extensive cemeteries in which urns were deposited with the ashes of the dead and grave goods [49]. Livestock was one of the fundamental pillars of the economy of these people, which explains the more than 400 granite sculptures representing bulls and pigs—popularly known as "verracos" (boars)—scattered throughout the *Vettonia* or region of the Vetton people (Figure 3) [50,51].

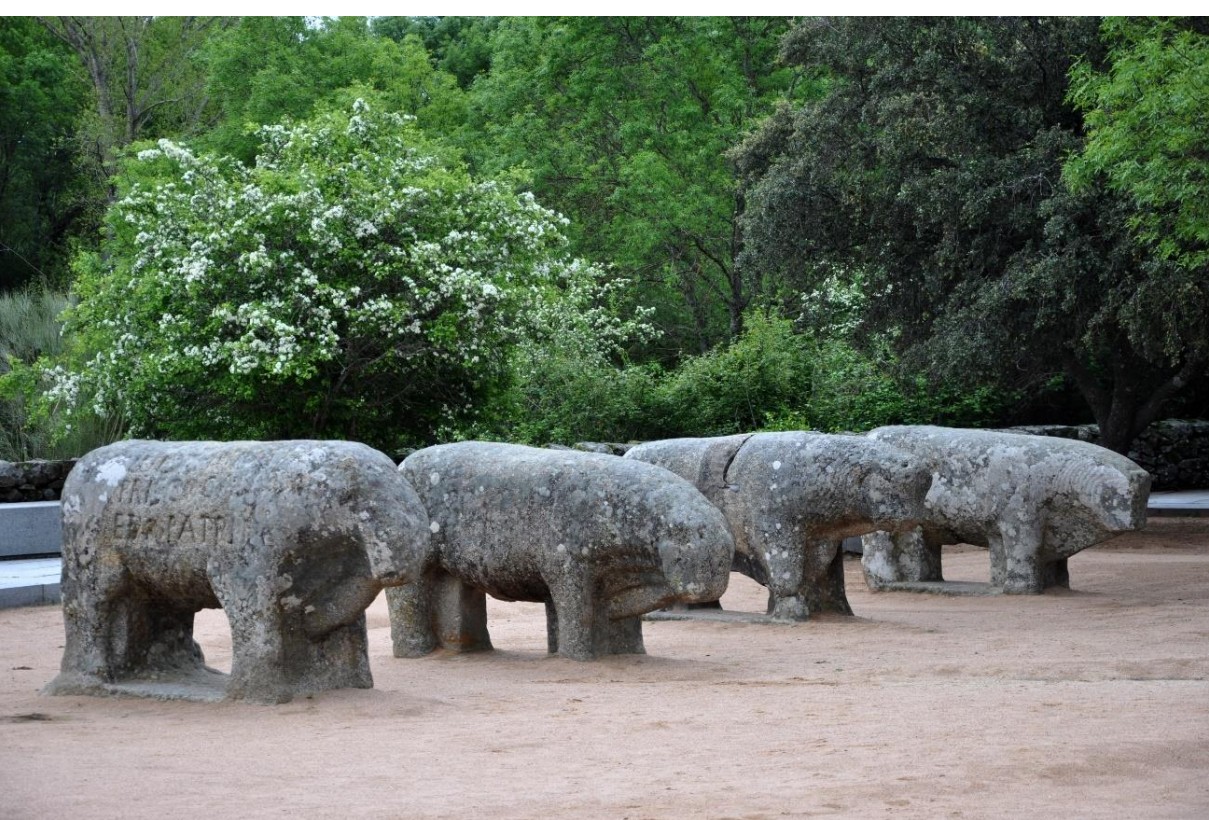

**Figure 3.** "Toros de Guisando" (El Tiemblo, Ávila), four Late Iron Age granite sculptures of more than 2.5 m in length representing bulls.

The Ulaca *oppidum* is located near Villaviciosa (Solosancho), a village about 30 km away from Ávila capital. At the end of the Iron Age (3rd—1st centuries BC), a community of the Vetton people settled on Ulaca hill, becoming the most important nucleus of the Amblés Valley. Its approximately 1500 inhabitants built a fortified settlement of around 70 ha, one of the largest known in Celtic Iberia [52,53]. Ulaca and its surroundings have been declared an Asset of Cultural Interest with the category of Archaeological Zone, the highest level of protection of historical heritage in Spain. Nowadays, the site has a guard, is fenced, and inside, cows and horses still take advantage of the rich pastures of the hill [54,55]. The itinerary to follow at the Ulaca site is marked by a series of information panels, but its access is only allowed on foot following a path on uneven terrain, with steep slopes that overcome a 250 m incline to the summit, located at about 1500 m of altitude.

Once the visitor reaches the highest part of the hill, it is possible to contemplate a landscape dominated by granite rocky areas, among which a series of stone structures erected by the ancient inhabitants of Ulaca are scattered (Figure 4). Among them, the following stand out: (a) the gates built in its more than 3 km of walls [56]; (b) the sanctuary, with its famous rock altar oriented towards the highest peaks of the mountain range "Sierra de la Paramera" [57]; (c) a ritual sauna semi-excavated in the rock [58]; (d) a couple of excavated and restored houses [52]; (e) impressive granite quarries [59]; and (f) a ruined building of great proportions, known as "El Torreón" (the Tower) [60,61], built with large granite blocks that differentiate it from the more than 250 common constructions that can be seen scattered around different parts of the *oppidum*. In addition, the archaeological excavations carried out at the site in the last two decades have yielded the location on the northern slope of the hill of a necropolis and an area of artisan workshops, although these elements are not open to visitors for now [62–64].

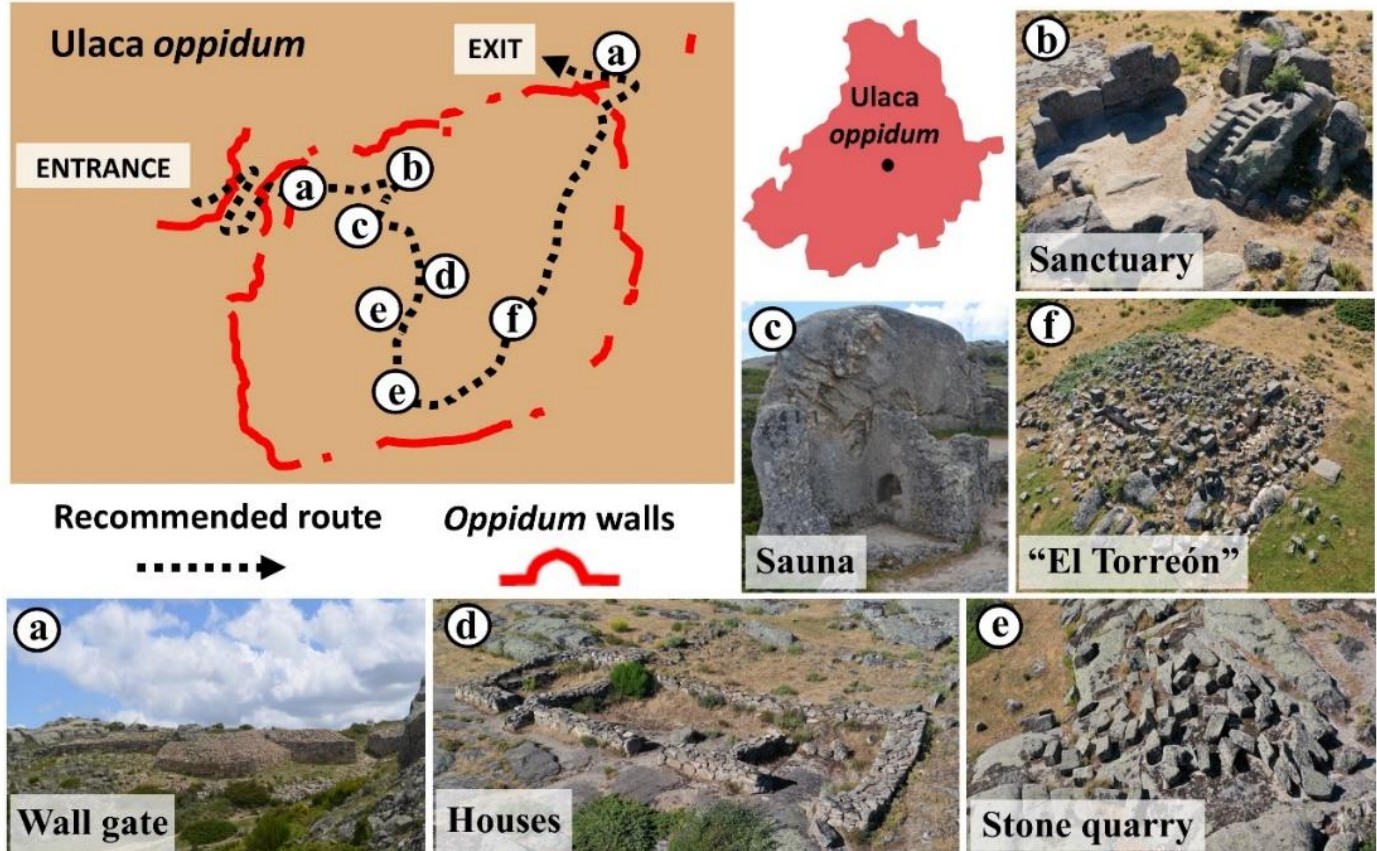

**Figure 4.** Principal monuments found within the Ulaca *oppidum*: (**a**) Wall gate, (**b**) Sanctuary, (**c**) Sauna, (**d**) Houses, (**e**) Stone quarry and (**f**) "El Torreón".

## 3. Materials and Methods

Virtual tours can be defined as immersive digital tools that allow visiting different spaces or places in a certain environment. The simulation of this environment is based on the interconnection of a series of 360° panoramic images, which allow the complete visualization of the space, generated by means of specific software. As a general rule, this virtual space is accessible from any computer or mobile device with internet access, providing a similar sensation to that of being on the site. In addition, information can be linked to these virtual tools to facilitate the user's understanding of the environment shown.

Below, the process of developing the ULACA <sup>VIRTUAL TOUR</sup> is presented.

### 3.1. Workflow for the Creation of the Virtual Tour

The ULACA <sup>VIRTUAL TOUR</sup> platform is essentially based on the use of 360° images capable of providing a sensation of complete immersivity at a reasonable cost. These images, displayed on the platform in spherical projection, have their own coordinate system. In this way, it is possible to position a button in coordinates (commonly called hotspots) and associate it with multi-source and multidisciplinary information in a specific position of a panorama through the angles θ and φ (Figure 5).

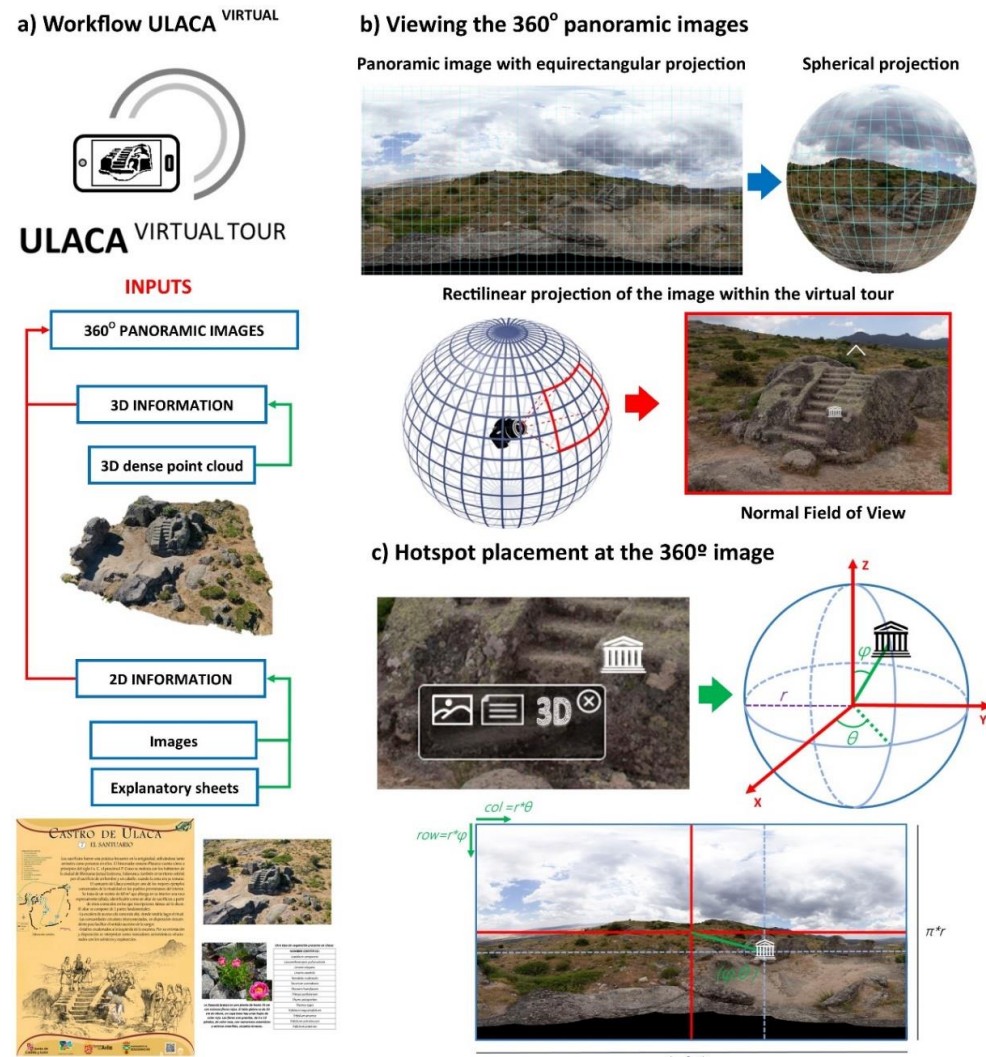

**Figure 5.** ULACA <sup>VIRTUAL TOUR</sup>: (**a**) Information about the virtual tour. (**b**) Visualization of the panoramas that integrate the virtual visit. (**c**) Positioning of the information in the panoramas.

### 3.2. Panoramic Images

After a detailed study of the different possibilities of capturing panoramic images, it was decided to use a reflex camera and fisheye lens, based on seven images (at 60° intervals and an overhead shot). The main advantage of using these devices instead of 360° cameras (which automatically generate a panoramic image), is the higher quality and control of parameters (ISO, exposure, white balance, etc.). In this work, a Nikon D5600 model reflex camera with a 10.5mm 1: 2.8G fisheye Nikkor lens, capable of capturing images with a resolution of 24 MPx, was used. On top of the above, a photographic tripod and a Manfrotto 303SPH panoramic head was required, to obtain precise photographs in angular positions every 60° (Figure 6). Before taking the photographs, it is necessary to calibrate the camera (together with the objective) on the anterior patella, in order to align the entrance pupil with the axis of rotation of the panoramic patella and thus eliminate parallax. It is also necessary to configure the internal parameters of the camera, such as the aperture of the diaphragm, the exposure time, or the ISO value of the sensor, together with the activation or not of the HDR mode (Figure 6). Once this is done, photographs can be taken adopting the shooting protocol already explained and shown in Figure 6. In this way, the previous photographs are processed in the Hugin open-source software (http://hugin.sourceforge.net/, accessed on 27 December 2021), and, according to the workflow shown in Figure 6, the panoramic photo is generated. The process starts with the key-point extraction and matching among individual images. The extraction of key-points was performed by using the well-known algorithm SIFT (Scale-invariant feature transform). Then, the matching was carried out by using an L2-norm and the RANSAC (Random Sample Consensus) algorithm as suggested by Brown and Lowe [65], allowing us to compute the homography matrix between consecutive cameras. After that, a global registration phase was carried out with the aim of minimizing the error accumulation. During this stage the distortion parameters of the camera, its focal length and angular positions were optimized. Finally, the panorama was generated by projecting the images on the spherical coordinate system. During this stage a gain compensation and multi-band blending were used in order to fuse all the images property from the radiometric point of view [65]. As a result, a total of nine panoramas were created. Each one of them was stored in .jpg format with a low compression ratio to avoid the presence of artifacts.

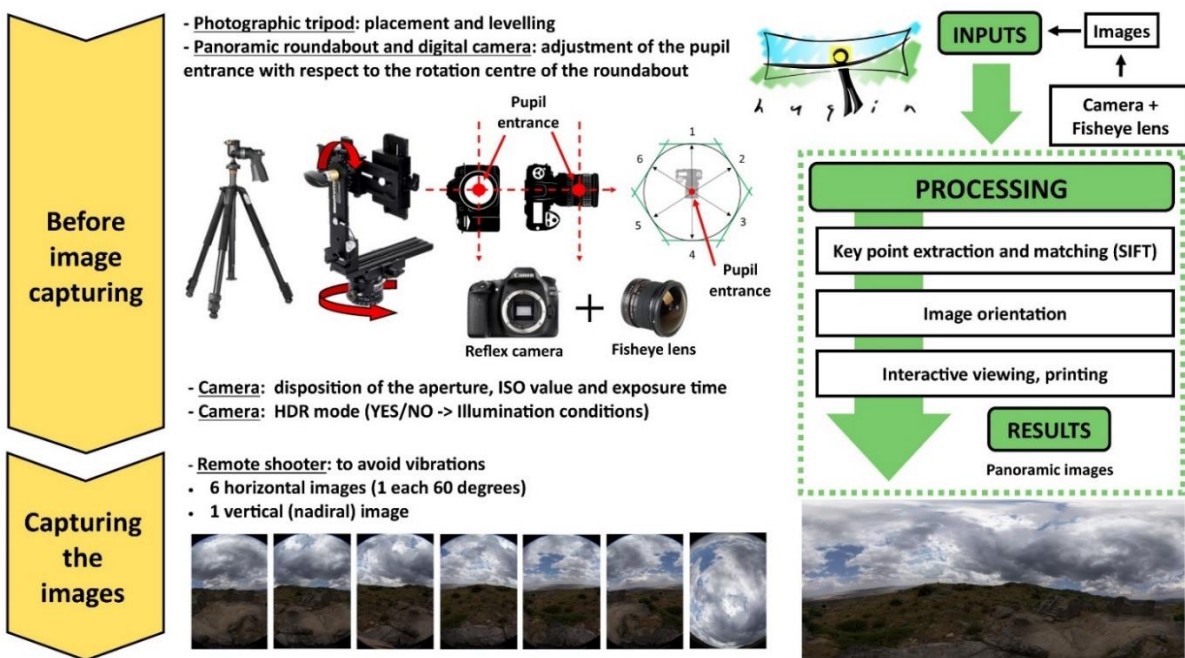

**Figure 6.** Protocol for data collection and processing of photographs to obtain panoramic images.

In addition, 56 high-resolution panoramic images (in .tif format) were totally generated: (i) of the main monuments of the Ulaca *oppidum* (Figure 4); (ii) of the locations in which there are information panels; (iii) of the most representative places of the archaeological site due to its landscape interest; (iv) of intermediate locations for the transition between the previous scenarios; (v) and finally, several panoramic images of the 2018 excavation campaign carried out in "El Torreón", in which the work dynamics of the entire archaeological team is shown.

### 3.3. 3D Documentation of Monuments Aerial Photogrammetry

In order to map and document the different monuments of the Ulaca *oppidum* (Figure 4), photogrammetric techniques were applied. These techniques allow the construction of cartographic products with 2D and 3D metric properties from aerial images collected from a drone. In the present research, a DJI Mavic 2 Pro (Figure 7 and Table 2) was used to capture the images of the different flights made over the monuments of the archaeological site, following an oblique/convergent photographic shooting protocol. Photographs were taken ensuring an adequate overlap among images (around 80–90%) and performing different passes varying the pitch angle of the drone by 10–15° and maintaining a constant distance to the monuments. For this task, a series of photographs were taken in a single flight following a circular sequence (360°), with the point of view of each image towards the center of the monument, flying in a circular way at three heights above the ground (15 m with a camera tilt of 60°, 30 m with a camera tilt of 45°, and 60 m with a camera tilt of 30°) (Figure 7a). The flight was then planned according to the drone model, the location and extent of the terrain, the orography, and the meteorological conditions, since Ulaca is located at a high altitude where strong winds are frequent. For flight planning, the author's software called MFLIP was used, which estimated a mean ground sampling distance (GSD) value of 1.5 cm [66]. In addition to this, several control points uniformly distributed throughout the Region of Interest (ROI) were obtained for the subsequent scaling of the photogrammetric model, with a GNSS (Global Navigation Satellite System) instrument (Topcon GR-5), with a ± 1 cm accuracy. These objectives were easily identifiable in each of the photographs when using target plates, allowing us to solve the external orientation and the georeferencing of the consequent analyses (Figure 7b).

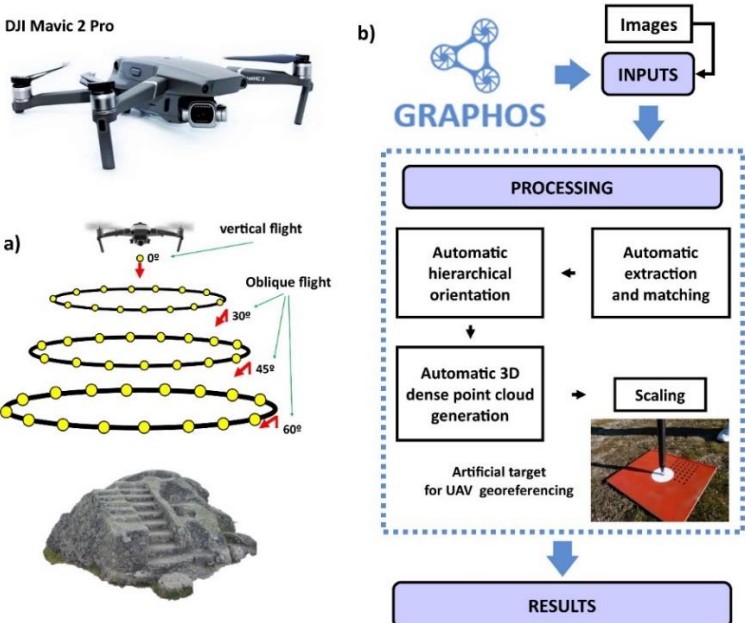

**Figure 7.** DJI Mavic 2 Pro. (**a**) Flight plan following an oblique/convergent photographic shooting protocol for the documentation of the monuments within the Ulaca *oppidum*, (**b**) Workflow for the photogrammetric processing of images when obtaining 3D point clouds.

**Table 2.** UAV sensor's specifications.

| DJI Mavic 2 Pro | |
|---|---|
| Autonomy | 31 min |
| Total weight | 907 gr |
| Maximum flight speed | 72 km/h |
| Accuracy of the on-board GPS | ±1.5 m (x,y) and ±0.5 m (z) |
| **DJI Mavic 2 Pro Camera** | |
| Sensor | 1″ CMOS<br>Effective pixels: 20 million<br>FOV: 77 |
| Objective | Format equivalent to 35 mm: 28 mm<br>Opening: f/2.8–f/11<br>Focus distance: 1 m to ∞ |

Once the different flights were made, images were processed with photogrammetric reconstruction software. After considering all the existing programs, the open-source tool GRAPHOS was selected [67].

Thanks to GRAPHOS, six dense point clouds of the monuments of the Ulaca *oppidum* (Figure 4) were generated (stored in .laz format). The metric control and final precision of the different point clouds obtained can be seen in Table 3. The Root Mean Square Error (RMSE) results from the residual errors at control points (considering in all cases, the use of eight control points in the form of target plates and with GNSS coordinates with ±1 cm accuracy. As explained previously, the target plates were uniformly distributed throughout the ROI, with the aim of a homogeneous distribution of errors).

**Table 3.** Metric control and final accuracy of the different point clouds.

| | Ground Sampling Distance (GSD) | Root Mean Square Error (RMSE) |
|---|---|---|
| Wall gate | 1.9 cm | 1.5 cm |
| Houses | 1.4 cm | 1.2 cm |
| Stone quarry | 1.5 cm | 1.1 cm |
| Sauna | 1.5 cm | 1.2 cm |
| Sanctuary | 1.5 cm | 1.3 cm |
| "El Torreón" | 1.5 cm | 1.2 cm |

*3.4. Information Required for Understanding the Archaeological Site*

An essential part of this kind of project is the correct compilation, homogenization, and standardization of the information for the understanding of the site under study. However, in these contexts, the information: (a) tends to come from different sources; (b) is usually in numerous places (dispersed depending on its owner, importance, and historical significance...); (c) or has different typology and format (heterogeneous information). Therefore, the search and compilation of information is frequently a complicated task, which must be carried out in an exhaustive transversal way [68]. It must be also considered that it is not always possible to find the required information and is convenient to establish search criteria and several preliminary objectives to estimate the resources to be invested in the development of this part of the research.

In the case of the Ulaca *oppidum*, the following information was compiled: (i) historical/archaeological; derived from the research team working at the site [52,62,63]; (ii) geological; consulting the website of the "Geolodía" outreach initiative, which took place in 2018 at the Ulaca *oppidum* (https://geolodiaavila.com/geolodia-avila-2018/, accessed on 27 December 2021) and promotes geological field excursions guided by specialists; (iii) floristics and fauna; obtained from different investigations in the area [69,70]. All these data were complemented with photographic material of elements of interest located in the Ulaca *oppidum* and its surroundings. Subsequently, all the information was rigorously

homogenized and standardized in different formats (texts and explanatory sheets). On top of the above, the Culture Territorial Service of the Junta de Castilla y León provided the project with all the informative posters present at the site, which we included to increase the interest and understanding of the visitors to the Ulaca *oppidum*.

In total, the following were generated: (i) five cards (in .png format) explaining and showing the action of humans still present in the landscape of Ulaca, such as: the pivot hole carved into the rock in which the wooden door would fit to close the northern access to the *oppidum*; the wagon wheel tracks visible in the rock; several remains of pottery; wedge-holes in the stone quarries for the extraction of blocks for the construction of the monuments; or one of the bull sculptures found in the surroundings of the *oppidum*; (ii) 17 cards (in .png format) where the flora and fauna that can be found in the site are described (4 fauna cards and 13 flora cards). It should be noted that the current inhabitants of the area continue to dedicate themselves to raising livestock, so it is very common today to find cows and horses grazing within the archaeological site; (iii) five sheets (in .png format) explaining and showing: the different characteristics of the granitic landscape (such as erosion and weathering of the rock, the existence of joints and fracture zones or the for-mation of rock basins); the presence of springs in the area; or information on the formation of the Amblés Valley; (iv) 14 images of the different explanatory panels of the *oppidum*; and (v) seven documents (in .pdf format) and images (in .png format) showing and explaining the uses and function of the main monuments of the Ulaca *oppidum* (Figure 4).

### *3.5. Architecture of the Virtual Platform*

The sources of information previously identified, as well as the 3D point clouds and panoramic images, were integrated in a unique virtual environment. This virtual environment uses the panoramic images as the main support to which is attached the different information sources. This environment was developed by using the low-cost software Pano2VR® (https://ggnome.com/pano2vr/, accessed on 27 December 2021). Within this software the virtual platform was programmed by considering different web-based languages, such as HTML (markup language), CSS (design language), and JavaScript (programming language). The HTML and CSS languages were used for designing the content of the platform as well as its graphical user interface (GUI) (Figure 8). This GUI was made up by a HTML header and body. The header is de-voted to show information related with the entities involved. Meanwhile, the body is used for projecting the panoramas, the information, the map of the site, and a navigation menu. The information of the platform is shown by means of hotspots which are HTML entities that include a JavaScript function that allow an image file to be shown when the user clicks on it. The navigation menu includes several JavaScript functions that allow the point of view to be changed by rotating the panorama. Additionally, this menu includes zoom functions, an icon for open/close the map as well as several icons for opening PDF files with general information such as a historical guide or a geological guide. These PDF files are rendered by using an iframe entity (HTML language).

It is worth mentioning that some of the hotspots include the possibility of rendering the 3D point cloud. This rendering was performed by using the open-source library Potree® (http://potree.org/, accessed on 27 December 2021). This platform is based on the WebGL technology and is capable of quickly rendering heavy point clouds, as well as incorporating metric operations, such as linear measurements, areas, or volumes, that may be of interest to professionals. The connection between this engine and the virtual tour was made up by using an HTML page on which is embedded the 3D point cloud. This viewer also includes several JavaScript functions that allow the coordinates of points, the value of linear measures or the area of a specific place to be obtained.

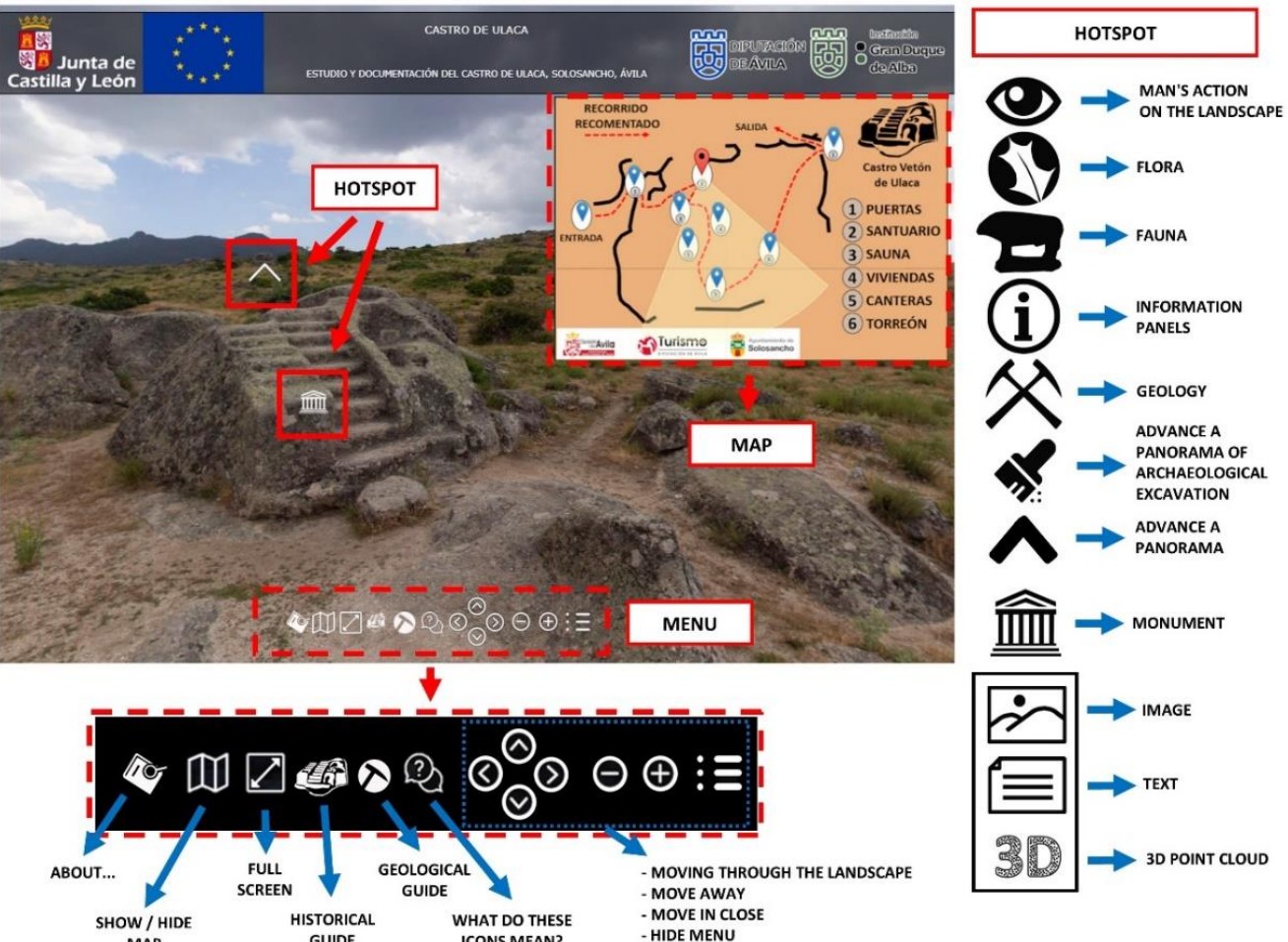

**Figure 8.** View of the ULACA <sup>VIRTUAL TOUR</sup> platform.

## 4. Results: ULACA <sup>VIRTUAL TOUR</sup>

The result of the virtual visit of the Ulaca *oppidum* can be consulted in the following link: http://tidop.usal.es/Ulaca/ (accessed on 27 December 2021).

For the creation of the virtual environment, made up by the 360° images, a total of 41 spherical panoramas were required. These panoramas were introduced into a virtual environment with the low-cost software Pano2VR (http://Pano2VR.com, accessed on 27 December 2021). The GUI of the platform was carried out using a specific author's plugin for Pano2VR as shown in the previous section (Figure 8). Figures 9 and 10 show the appearance of the platform when the user consults different types of information sources such as images, PDF files, and 3D point clouds.

The integration of the different sources of information and 3D point clouds in the 360° images required the creation of a GUI constituted by several menus and toolbars with different functions (so-called hotspots), a map, and a menu focused on providing the best user experience (Figure 8). An additional programmed function allows the panorama to rotate when the user stops interacting with the platform for a few seconds. In this way, it is possible to have a continuous view at a constant rate of the landscape present in the panorama.

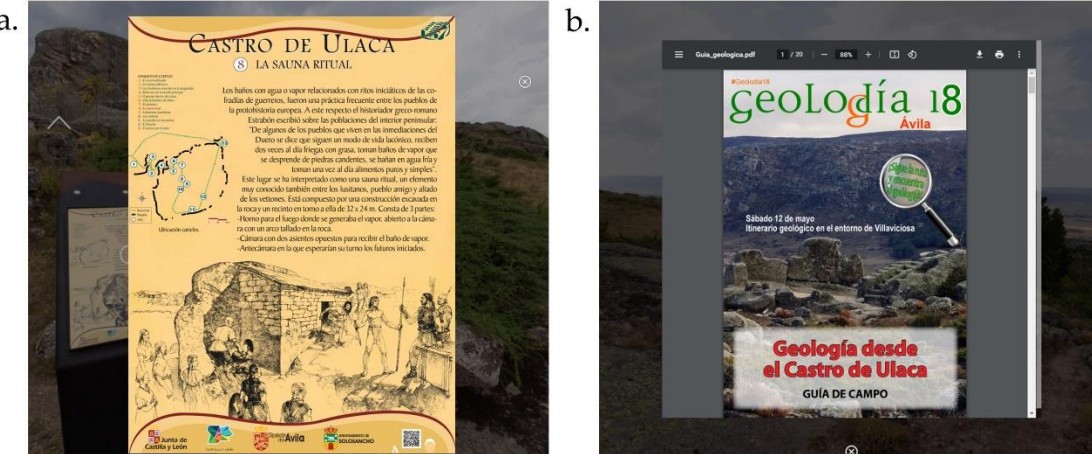

**Figure 9.** Virtual tour: (**a**). Appearance when the user clicks on a hotspot with information related to the explanatory panels; (**b**). Appearance when the user clicks on the hotspot devoted to the geological guide (navigation menu).

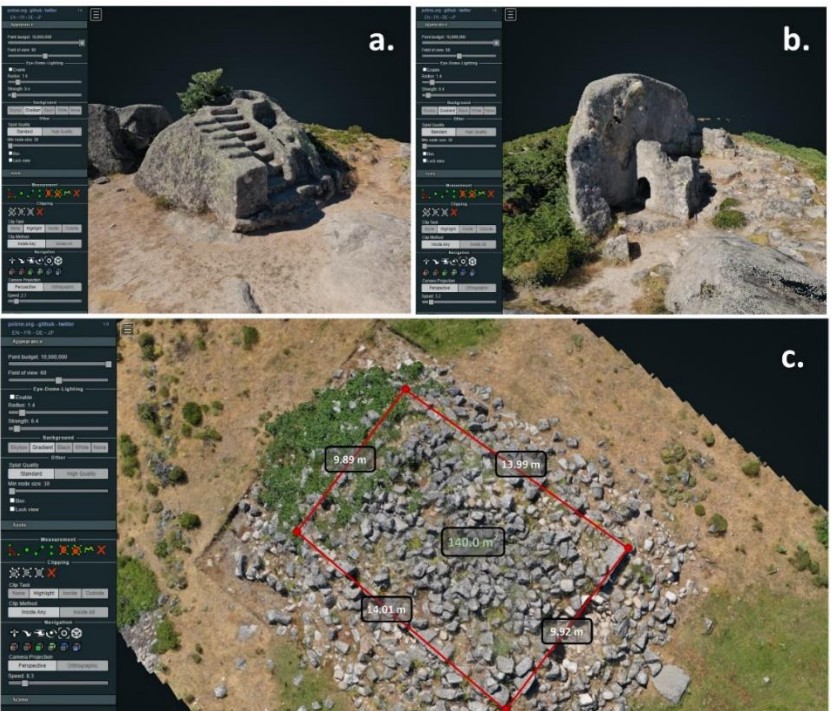

**Figure 10.** Different point clouds of the monuments of Ulaca *oppidum*: (**a**). Altar; (**b**). Sauna; (**c**). "El Torreón", in which it is possible to observe the way of measuring and calculating volumes.

## 5. Discussion Virtual Tours: Challenges and Possibilities

The main challenge of a virtual tour is, as stated by Grima [71] (p. 82), the existence of a whole series of experiences and attributes that can be observed on a trip to an archaeological site, which, however, cannot be transmitted through the digital world. In the case of the virtual tour of Ulaca, it is not easy to transmit the difficult access to the site, its enormous size, the potentially adverse weather, the intense smell of the plants, the almost absolute silence, or the luminosity of the starry sky. However, we are sure that in the future technological advances will alleviate, at least partially, these inconveniences. In addition, other challenges to overcome when developing this virtual tour are: (i) the difficulty of coherently integrating all data from the different fields of science, engineering, and humanities; (ii) the creation of an attractive platform design for the user; (iii) the

selection of the most appropriate formats to display the information on the platform; (iv) to correct the imbalances among the panorama, the map, the menus, the information to be displayed, the logos, and the texts, so that they look balanced on the different devices (whether they are computers, smartphones, or tablets); and (v) the detection of systematic failures and their subsequent debugging.

In any case, virtual visits are of enormous interest due to the multiple possibilities they offer to potential users, as is exemplified in the case presented here.

According to different surveys [54], visitors interested in the Ulaca *oppidum* are fundamentally non-expert adult users. This group is mainly made up of national tourists or inhabitants of the province of Ávila. As a rule, these visitors do not demand technical information about the *oppidum*, but rather general information about the site, to easily understand its history and peculiarities. Thus, they are usually interested in aspects related to (i) the inhabitants of the Ulaca *oppidum*, the Celtic, and Vettonian worlds; (ii) the characteristics of the Ulaca constructions; (iii) its environment (geology, landscape...); and (iv) the principal fauna and flora of the place. For this reason, information on the archaeology, geology, and ecology of the site and its surroundings was incorporated into the virtual visit, to perform a comprehensive approximation to the landscape of Ulaca and the Amblés Valley. This valley, of over 40 kilometres long and 10 km wide, was occupied uninterruptedly since at least the Neolithic–Chalcolithic period (IV–III millennium cal BC) [72] and fits perfectly with the current conception of cultural landscape, "*the result of people interacting over time with the natural medium, whose expression is a territory perceived and valued for its cultural qualities, the result of a process and the bedrock of a community's identity*" [73] (p. 21). In order to provide the user with a better visualization of this cultural landscape, two features were implemented in the virtual visit to appreciate the spatial processes that reflect the continuous action of the human being over the natural environment: (i) a hotspot of elements of interest that shows how human beings have left their mark on the landscape, such as the traces of cartwheels; the wedge-holes in the rock due to the extraction process in the quarries; or the ceramics found in some parts of the site; (ii) a constant automatic rotation of the panoramas (when the user does not interact with the virtual tour for 40 seconds) that allows a general visualization of the landscape. These two aspects make this interactive tool a great help to understand how humans have adapted to the environment and shaped the landscape over time.

On the other hand, the visibility of the archaeological remains in Ulaca is an ideal aspect for the investigation and visit by the public, but also entails their deterioration due to the exposure to inclement weather during the last two millennia. This is especially evident in the sanctuary and the sauna, which also endure the wear and tear caused by visitors who climb these structures daily and on special dates (during the massive celebration of the Celtic Moon festival). For this reason, the 3D models of these two monuments generated for the virtual tour are considerably interesting for having a reliable image of their current state, which can be compared with their future evolution to, if necessary, plan the corresponding protection and conservation actions.

In this virtual tour, images of the archaeological excavation carried out in "El Torreón" during 2018 were incorporated. This aspect is fundamental so that the visitor's experience is not limited solely and exclusively to contemplation, but rather an approximation (albeit shallow) of the process associated with the excavation works, which, in addition, should be one of the bases to promote tourism practice based on archaeological heritage. Such tourist activity, properly planned and managed, can represent an opportunity in "unfavorable" areas (depopulation of the rural environment—the Autonomous Community of Castilla y León, and more specifically the provinces of Soria, Ávila, Segovia, Zamora, and Palencia are rural territories affected by severe demographic issues, that is, areas with population densities of 12.5 inhabitants per square kilometer or less, or those that have lost an average of at least 1% per year of the population in the period from 2007 to 2017) or "under the consequences of a disaster" (Navalacruz forest fire, 2021—On 14th August 2021, a forest fire started in Navalacruz (Ávila) that ended up seriously affecting the Ulaca *oppidum* and

all its surrounding landscape. The fire devastated more than 22,000 hectares, becoming the most important forest fire in recent years in Spain), as is the case in Ulaca. In fact, there are examples, such as the initiative promoted by Lorcatur after the 2011 earthquakes in Lorca (Murcia), whose program "Lorca, open for restoration" was a boost for the city and was an idea awarded by the World Travel Market (WTM) in the 2012 edition of the Global Awards. It is, therefore, a matter of turning a priori weakness into an opportunity and tools such as the one presented in this work can be key in the generation of these types of dynamic tourism products.

One of the most innovative uses of this platform is possibly to promote better information management to facilitate the investigation of the teams (multidisciplinary) in the excavation and study processes of archaeological sites. The study of an archaeological site is frequently associated with a large amount of documentation: bibliographic sources, specifically of the archaeological excavations, data analysis, administration, oral sources, graphic documents (plans, photographs...), among other sources of information that may be interesting, related to geology, ecology, etc. The combination and management of all this information sometimes becomes complex, making investigations difficult. The implementation of this platform in an inclusive environment (thanks to the 360° visualization of different locations in the Ulaca *oppidum*) and its possible use as a geospatial database (allowing information to be in a specific point of the panorama) facilitates the link and association between graphic data (2D/3D visualization) and historical, geological, ecological documentation among other data of interest, allowing the development of investigations in an agile way. Thus, the global environment of the area or element under study, with geolocated information is available. In addition, it is worth highlighting the novelty of including 3D information in tools with these characteristics. The 3D documentation of an element or environment is digital metric information that allows for useful formal analyses to understand and study the reconstructed scene. Measurements in buildings, such as those obtained in "El Torreón" from this site, allow the construction of new hypotheses around its use and functionality [60,61]. This third dimension allows an improved understanding of the environment, as can be seen in the reconstruction of the northwest area of the *oppidum* walls, where the three lines of defenses can be perfectly observed.

For some authors, the global life cycle of Cultural Heritage involves four phases: knowledge, use, communication, and management [74]. In the case of the present platform, which allows the storing and management of data in different formats and from different sources, all the information is available to the stakeholders [75]. It should be also noted that most heritage files and property records do not contain 3D information [76], an aspect that could be partially covered with the workflow proposed here.

The role of this platform for didactic use in formal and informal contexts is also clear, since it can be a teaching/learning tool at various levels of higher education, such as Compulsory Secondary Education or university studies. Virtual tours have been used in education since the late 20th century. As an example, its application in pedagogy is reflected in the investigations of Foley [77], which develops the possibilities of bringing students closer to different places and museums and being able to interact with them through images or videos. Procter [78] highlighted the possibility of using multi-user virtual environments (MUVE) that allow the interaction of students with the virtual space, viewing objects or simulations generated for the didactics of a place.

In Compulsory Secondary Education and university studies, Ulaca Virtual Tour platform meets several of the competencies that students must acquire at these educational levels, mainly through Information and Communication Technologies (ICT). In fact, ICTs are considered one the basic competences that students must possess at the end of compulsory education in the Spanish educational system.

Regarding the university environment, this platform can be used in numerous university degrees as teaching support in subjects related to Archaeology, History, Heritage, Architecture, Geology, Biology ... , allowing an eminently practical teaching system, compared to the traditional passive, theoretical, and repetitive learning. The experience of

meaningful learning thus displaces the mere assimilation of knowledge in fixed compartments. It also enables the reinforcement of skills acquired in previous levels of training. Likewise, this tool can be especially useful in times of pandemic, as an alternative to field trips, both for schoolchildren and for high school/university students. In addition, virtual tours are a resource that favors the integration of students with economic or mobility difficulties since it allows the enjoyment and benefits of the tour [79] without assuming the economic cost of the trip or the poor accessibility of some sites. These tools are also a particularly useful resource for retired university students (University of Experience or University for the Elderly), who are increasingly present in the university context [80] (p. 244). In this way, it becomes a very useful tool in the implementation of educational innovation methodologies.

In addition, the use of this tool in the teaching/learning system perfectly fits with the STEAM (Science, Technology, Engineering, Arts and Mathematics) concept that promotes a new integrated education initiative of Science, Mathematics and Technology (in general, not only computer science) with two well differentiated characteristics: (i) teaching–learning of Sciences, Technology, Engineering, Arts and Mathematics in an integrated way, rather than as compartmentalized areas of knowledge; (ii) with an Engineering approach regarding the development of theoretical knowledge for its subsequent practical application, always focused on solving technological aspects. STEAM initiative thus improves the explicit assimilation of concepts from different disciplines or skills related to the use and development of technologies, among many other positive impacts on students.

Likewise, the possibility of using these digital tools in historical/archaeological museums cannot be overlooked. The visualization of the virtual tour of an archaeological site in 2D (through computers) or in 3D (through computers with the help of different glasses and helmets—Virtual Reality) next to the collection showcases can provide the necessary context to better know and understand the place where the different exhibited objects appeared. Similarly, and to enrich the use of these tools in the presentation of the archaeological sites, it may be interesting to incorporate into the virtual tour a 3D reconstruction of the objects deposited in museum collections [81]. This aspect is especially interesting, since sometimes the archaeological objects discovered in a site are distributed in different institutions. In this way, it would be possible to simulate a virtual museum of all the objects belonging to that archaeological site, contextualized in a spatial and temporal way. It could be very attractive for visitors to exploit the use of Augmented Reality (AR) techniques, which allow virtual objects to be placed in the real world, and Mixed Reality (MR), which takes AR one step further and allows virtual and real objects to merge and interact with each other. All this will allow a better exploration of the cultural and social meaning of these archaeological objects and the culture or historical period to which they belong, as well as a better understanding of their past use or function, greatly improving their visibility to a non-expert public. A complete technological innovation that, with the help of smartphones and tablets, promises to revolutionize visits to archaeological sites and museums.

Finally, virtual tours can play a fundamental role in the inclusion of "non-publics" [82], that is, groups generally excluded from archaeology, such as, people with motor difficulties who want to visit a site with difficult access [83], as is the case of Ulaca.

## 6. Conclusions

In this work a virtual tour made on the Ulaca *oppidum*, ULACA ᵛᴵᴿᵀᵁᴬᴸ ᵀᴼᵁᴿ, is presented. There is no doubt that these types of heritage interpretation tool, for educational use in formal and informal contexts or for the dissemination of cultural heritage, are widely implemented. However, the novelty of this research is the possibility of using this tool for other purposes, such as: (i) unifying information from the different fields of science, since not only purely archaeological/historical data are included, but also information of geological or ecological nature; or (ii) the storage and management of information (by enabling its use as a geospatial database) in excavation contexts and

archaeological or environmental studies, facilitating the investigation of experts by allowing the pooling of information from different sources and formats (photographs, texts, 3D models of geographic space...), being especially novel in this area its integration as a tool for the study of cultural landscapes. In this way, the combination of 360° images with immersive capacity together with Digital Terrain Models (DTM), Orthophotography, 3D Graphics Representations and other conveniently assembled cartographic products allows the observer to recreate the space with clarity while responding quickly to the demand of basic information about the place. This type of visual impact not only increases the visit time on the web, but also encourages the virtual traveler to the realization of a real cultural and/or tourist trip. Thus, this virtual tour is considered as a basic guide to good practices, due to the information it contains and its precision, in relation to the time taken for its development.

These types of applications constitute a tool to diversify and enrich the tourist offer of the region of Ávila. There is no doubt that the province is attractive as a destination for rural and nature tourism, ranking in this regard among the ten most relevant destinations in Spain. However, requirements are increasingly demanding, in addition to the natural values and quality of the infrastructures and tourist facilities. In this sense, it is necessary to support non-massive practices, consistent with the fragility of the territories, such as archaeotourism, the object of this work, ornitourism, or astrotourism. In Ulaca, light pollution is minimal, so the visitor can enjoy the vision of an incredible starry sky. In addition, recent archaeoastronomical studies have revealed that the alignment of the altar with the highest peaks of the mountain range "Sierra de la Paramera" is not accidental, since it would be related to the height of the sun, as it passes through the summit known as the "Risco del Sol" (Crag of the Sun), during the winter solstice [57]. Apart from the activities carried out annually to attract visitors to Ulaca and other *oppida* in the province of Ávila, such as the Celtic Moon or other "Celtic" markets, this tool can serve to claim and attract, motivate, and seduce an audience interested in other types of themes, beyond the purely archaeological aspect. Consequently, this type of dynamic and open tools can provide great added value by combining different innovative elements, presenting a great journey and future, with the common denominator of the use of new technologies.

It should be finally noted that the possibility of integrating multi-source information can benefit and facilitate training and education actions for citizens. STEAM teaching/learning in an integrated way, instead of as compartmentalized areas of knowledge, improves the explicit assimilation of concepts from different disciplines, skills related to the use and development of technologies, among many other benefits that have a very positive impact on students. If these tools use cultural heritage as a backdrop, they can be beneficial by allowing the acquisition of knowledge and skills that help them explore and understand the importance of heritage and its environment [84]. Individuals will be able to act consistently in an active, inclusive, resilient, responsible, critical, and supportive way, in accordance with the identity values of the society (avoiding social detachment, depopulation, and other current challenges), which are so closely linked to cultural heritage.

**Author Contributions:** Conceptualization, M.Á.M.-G., J.R.-H., C.S.B., G.R.Z., and J.R.Á.-S.; methodology, M.Á.M.-G., J.R.-H., C.S.B., and L.J.S.-A.; validation, M.Á.M.-G., J.R.-H., C.S.B., D.G.-A., M.P., L.J.S.-A., G.R.Z., and J.R.Á.-S.; formal analysis, M.Á.M.-G., J.R.-H., C.S.B., D.G.-A., M.P., G.R.Z., and J.R.Á.-S.; investigation, M.Á.M.-G., C.S.B., J.R.-H., D.G.-A., M.P., G.R.Z., and J.R.Á.-S.; resources, M.Á.M.-G., C.S.B., J.R.-H., J.F.H., L.T.T., J.F.F.G., T.R.H.T., L.J.S.-A., D.G.-A., C.D.-S., M.P., G.R.Z., and J.R.Á.-S.; data curation, M.Á.M.-G., C.S.B., J.F.H., L.T.T., J.F.F.G., D.G.-A., and M.P.; writing—original draft preparation, M.Á.M.-G., C.S.B., J.R.-H., L.T.T., T.R.H.T., D.G.-A., M.P., C.D.-S., G.R.Z., and J.R.Á.-S.; writing—review and editing, M.Á.M.-G., C.S.B., J.R.-H., L.T.T., T.R.H.T., D.G.-A., M.P., G.R.Z., and J.R.Á.-S.; supervision, M.Á.M.-G., C.S.B., and J.R.-H.; project administration, M.Á.M.-G., C.S.B., J.R.-H., D.G.-A., M.P., G.R.Z., and J.R.Á.-S.; funding acquisition, M.Á.M.-G., C.S.B., J.R.-H., D.G.-A., M.P., J.F.F.G., G.R.Z., and J.R.Á.-S. All authors have read and agreed to the published version of the manuscript.

**Funding:** This work has been partially funded by: European Union—Marie Skłodowska-Curie Individual Fellowships, H2020-MSCA-IF-2019 (grant agreement ID: 894785; AVATAR project); Government of Spain (European Project PCIN-2015-022); Government of Spain (National Project HAR2015-65994-R); Comunidad de Madrid "Programa sobre paisaje y patrimonio, LABPA-CM (H2019/HUM-5692). Programa de Actividades de I+D en CCSS y Humanidades de la Comunidad de Madrid"; Diputación Provincial de Ávila; Institución Gran Duque de Alba (Ayudas a la investigación sobre temas abulenses, convocatoria 2019).

**Data Availability Statement:** Data sharing not applicable.

**Acknowledgments:** The authors would like to thank the TIDOP Research Group of the Department of Cartographic and Land Engineering of the Higher Polytechnic School of Ávila (University of Salamanca) and the GESyP Group (Universidad Politécnica de Madrid) for allowing us to use their tools and facilities. In addition, we would like to thank the Junta de Castilla y León for providing information regarding visitors to the *oppida* in the province of Ávila. This work has been carried out within the framework of research projects funded by: the European Union through a postdoctoral fellowship to one of the authors within the actions of Marie Skłodowska-Curie Individual Fellowships, H2020-MSCA-IF-2019 (grant agreement ID: 894785; AVATAR project "Application of Virtual Anastylosis Techniques for Architectural Research" (http://avatar.polito.it/, accessed on 27 December 2021); to the Government of Spain through the European Project PCIN-2015-022, "Resituating Europe's first towns: a case study in enhancing knowledge transfer and developing sustainable management of cultural landscapes" and to the National Project HAR2015-65994-R, "Vettones: urbanism and society through non-destructive techniques"; to the Community of Madrid, through the program on landscape and heritage, "LABPA-CM (H2019/HUM-5692). Programa de Actividades de I+D en CCSS y Humanidades de la Comunidad de Madrid"; to the Provincial Council of Ávila through funds for archaeological works and to the Gran Duque de Alba Institution (dependent on the Provincial Council of Ávila) through the "Ayudas a la investigación sobre temas abulenses, convocatoria 2019".

**Conflicts of Interest:** The authors declare no conflict of interest.

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
