# Peer review of "Challenges and Possibilities of Archaeological Sites Virtual Tours: The Ulaca Oppidum (Central Spain) as a Case Study"

_remotesensing, doi:10.3390/rs14030524_

Round 1
Reviewer 1 Report
The presented article contains a very interesting topic of the integration of data from many sources of development, not only of an interactive database for cultural heritage objects, but also of creating a platform for virtual tours of a historical object. These types of studies are a modern and very good tool for getting to know cultural heritage objects for many environments. The article is presented in an interesting form as a case study for The Ulaca Oppidum.
At the beginning, the article contains a very broad introduction and historical characteristics of Ulaca oppidum and the Celtic archaeotourism in the region of Ávila and the dynamics of tourist visits in this region. This is undoubtedly interesting information from the historical, social or tourist point of view, but too extensive for this article. In my opinion, such detailed analyzes of the dynamics of visits in particular years or months are not needed (Fig. 2 and Fig. 3).
The basic question to ask is what is the purpose of this article? Do we present a developed virtual tour platform for a selected historical region and the technical aspects that have been solved? Do we justify the necessity of this study and its advantages?
Analyzing the content of the article, I conclude that this is the answer to the second question.
The section on materials and methods is the shortest section in the article and is described very briefly. It is good that there are figures 6, 7 and 8 which illustrate the adopted methodology of integrating individual data. Unfortunately, the methodology is too general. Only the hardware and software used are listed. It is not known what accuracy was achieved for individual products, what accuracy was obtained, what formats were used and, most importantly, what is the architecture of the built platform. Only the menu of the developed platform is shown.
The remainder of the article is a discussion on the possibilities of the developed platform and its advantages, as well as focusing on specific social groups. It is a very good text, but is it a scientific discussion on the adopted methodology for developing this platform?
So I go back to my question? What is the purpose of the article. If you provide general information about the developed platform and its capabilities, this is a good article
If this is a presentation of a multi-source data integration methodology? and the issues of creating an appropriate platform structure for the development of virtual tours for cultural heritage sites, taking into account many technical and methodological aspects, the article should be improved. Chapter 3 (materials and methods) should especially be edited. And the discussion should include a discussion of the problems that have been solved while creating this platform.
Author Response
Authors thank this referee for the high quality of this revision and all the hints and remarks identified. Manuscript has been reviewed and improved following your advices and hints. All your considerations have been added to the new version of the manuscript (see attached pdf).

Reviewer 2 Report
The overall research design is appropriate and the work is easy to read. The introduction is well structured, a sufficient number of references is present and the overall methodological framework is well described.
I will suggest adding some missing information in section 3.3:
-UAV sensors's specifications (Sensor type, size, pixel size, focal length)
-More details about the photogrammetric processing (metric control and final accuracy of the results)
Finally, in the conclusion, you report the possibility of using the ULACA VIRTUAL TOUR as a geospatial database. From what I've seen in the potree viewer the 3D models seem to be referred to a local coordinate system. Is that correct? are you planning a solution to these issues?
Author Response

(The authors gave the same response as above.)
